# The Use of Larval Sea Stars and Sea Urchins in the Discovery of Shared Mechanisms of Metazoan Whole-Body Regeneration

**DOI:** 10.3390/genes12071063

**Published:** 2021-07-13

**Authors:** Andrew Wolff, Veronica Hinman

**Affiliations:** 1Department of Biological Sciences, University of Maryland, Baltimore County, 1000 Hilltop Circle, Baltimore, MD 21250, USA; awolff@umbc.edu; 2Department of Biological Sciences, Carnegie Mellon University, 4400 Fifth Avenue, Pittsburgh, PA 15213, USA

**Keywords:** echinoderm, whole-body regeneration, larval

## Abstract

The ability to regenerate is scattered among the metazoan tree of life. Further still, regenerative capacity varies widely within these specific organisms. Numerous organisms, all with different regenerative capabilities, have been studied at length and key similarities and disparities in how regeneration occurs have been identified. In order to get a better grasp on understanding regeneration as a whole, we must search for new models that are capable of extensive regeneration, as well as those that have been under sampled in the literature. As invertebrate deuterostomes, echinoderms fit both of these requirements. Multiple members regenerate various tissue types at all life stages, including examples of whole-body regeneration. Interrogations in two highly studied echinoderms, the sea urchin and the sea star, have provided knowledge of tissue and whole-body regeneration at various life stages. Work has begun to examine regeneration in echinoderm larvae, a potential new system for understanding regenerative mechanisms in a basal deuterostome. Here, we review the ways these two animals’ larvae have been utilized as a model of regeneration.

## 1. Introduction

Tissue regeneration remains one of biology’s greatest mysteries. It is a complex process, involving the coordination of signaling cascades and gene expression programs, but this capacity to regenerate is not universally shared among animals. Even within animals capable of regeneration, the extent of regeneration varies depending on the tissue type and developmental stage [1].

Decades of research has identified general underlying principles of the processes of regeneration through the studies of phylogenetically diverse regenerative organisms. Regeneration begins with a response to a wound, creating a signaling center that instructs the remaining tissue to heal and replace lost tissue [2,3,4]. Often, progenitor cells, which are usually either resident stem cell populations or other cells that have become dedifferentiated following the injury, are recruited to proliferate to provide the cellular material for tissue restoration [5,6,7]. This is not always the case, as the cnidarian *Hydra* can achieve regeneration with limited contribution from the proliferation of new cells [8]. Whole-body regeneration also requires the ability of the animal to reset its major body axes through positional cues, such that the correct tissues are regenerated in the correct spatial domain [9,10].

While some understanding of the mechanisms of regeneration has been achieved, major questions remain unresolved. Perhaps the most important concerns the evolutionary history of regeneration and the homology of mechanism across the animal tree of life [11,12,13,14,15]. Viewing animal regenerative ability across a phylogenetic tree reveals no clear pattern in either regenerative capacity or mechanism, but as little is known about the array of species that it is hard to understand whether this complex process has evolved independently multiple times or has a common evolutionary origin of regeneration. Thus, it remains an open question whether regeneration is an ancestral feature of animal biology that was lost in certain lineages, or an adaptive trait that has been independently gained over evolutionary time. In an attempt to answer these questions of regenerative biology, research has begun to study regeneration in under sampled phyla. This will allow for the further discovery of regenerative mechanisms and help lead towards understanding how regeneration could be initiated in those lineages incapable of achieving this feat.

Echinoderms (phylum Echinodermata) are marine invertebrate animals found throughout the world’s oceans. There are five extant classes of echinoderms, including sea urchins (Echinoidea), sea stars (Asteroidea), brittle stars (Ophiuroidea), sea cucumbers (Holothuroidea), and sea lilies (Crinoidea). Most species undergo a biphasic life cycle, beginning with a free-swimming feeding larva that settles to the sea floor and metamorphoses into an adult animal. A notable feature of most adult echinoderm body plans is their radial symmetry, most commonly pentaradial. Echinoderms have traditionally been utilized as models of embryonic development [16,17,18,19]. Manipulations of sea urchin embryos have been performed to uncover information about the function of the nucleus as well as general embryonic development [20]. The gene regulatory network, the wired interaction of regulatory gene transcription, cis-regulatory modules, and signaling pathways, that drives the formation of the sea urchin larval skeleton remains the most complete of any cell type [21]. Echinoderms are of particular interest in evo-devo studies, as they are invertebrate deuterostomes and thus sister taxa to chordates. The sequence of the purple sea urchin genome (*Strongylocentrotus purpuratus*) has highlighted key features of echinoderm genomic structure and content in relation to other deuterostomes [22]. Regarding gene content, a reciprocal BLAST found that of roughly 29,000 sea urchin sequences identified, 7077 sea urchin proteins match a human protein and a similar number of mouse proteins [23]. Additionally, echinoderms share syntenic clusters with amniote vertebrates, including the pharyngeal cluster [24] and the ParaHox cluster in the sea star genome [25]. This phylogenetic placement allows for echinoderms to be used for elucidating ancestral deuterostome biological features. Beyond embryogenesis, echinoderms do not show typical signs of aging [26], also making them ideal for the study of senescence and cancer [27].

In addition to their amenities towards the study of development and evolution, echinoderms possess extensive regenerative capabilities. Members of each echinoderm class possess the ability to regrow lost adult body parts [28,29,30,31,32,33]. At each life stage, some echinoderms even undergo whole-body regeneration. Echinoderm larvae also exhibit fantastic regenerative capacity [34]. Because the larvae are developmentally proximal, possessing a body plan and cell types that are distinct from the adult form, echinoderms can be used to understand the relationship between development and regeneration. Echinoderms provide a unique phylogenetic position towards understanding the evolution of whole-body regeneration within the deuterostome clade. While their ability to regenerate is well known, the mechanisms underlying this regenerative capacity are still poorly understood.

In this review, we examine the advances echinoderms have provided in the study of tissue regeneration. More specifically, we will be discussing the regenerative mechanisms gleaned from two echinoderm classes, sea stars and sea urchins, during their larval stage. Additionally, due to their unique biology, we will also present ways these animals are used in studying whole-body regeneration.

## 2. Basic Biology of Larval Echinoderms

Being mainly benthic, many adult echinoderms are broadcast spawners, releasing their gametes into the water column for external fertilization. Once fertilized, echinoderm embryos undergo stereotypic deuterostome cleavage to form a blastula [35]. Gastrulation begins with the formation of the blastopore, ultimately becoming the anus. The three germ layers then become specified to form the necessary larval structures.

Larval echinoderms have a relatively simple anatomy (Figure 1). To aid in motility and the procurement of food, larvae possess ciliary bands composed of densely-packed ciliated cells. Sea star and urchin larvae also have a tripartite digestive system running along the anterior-posterior (AP) axis. Adjacent to this are the coelomic pouches, which house the larval germ line. It is from here that the rudiment forms, which allows for the process of metamorphosis to occur. Echinoderm larvae, while simple, have an array of neural subtypes, located throughout the larval body [36,37]. Additionally, sea urchin and star larvae mesenchymal cells perform a variety of functions, including immune function [38,39]. While the aforementioned larval structures and cell types are shared among sea urchins and sea stars, sea urchin larvae possess an endoskeleton derived from primary mesenchyme cells (PMCs) which sea star larvae lack [40]. Taken together, these echinoderm larvae are more complex than meets the eye, containing several differentiated cell types and structures that must be reformed during regeneration [41].

## 3. Advantages of Using Echinoderm Larvae Compared to Their Adult Counterparts

Throughout the last few decades, the primary studies on echinoderm regeneration have been on the reformation of adult tissue, and in the case of sea stars, adult arm regeneration. While studying adult echinoderm regeneration can inform us about deuterostome whole-body regeneration, it possesses several limitations. First, although adult sea stars undergo amazing regenerative feats, adult sea urchins cannot undergo whole-body regeneration, as the extent of their abilities is limited to their spines and tube feet [42]. Furthermore, while echinoderm larvae use homologous patterning mechanisms in development [40,43], these mechanisms are relatively unknown in the adult animals, with some exceptions [44,45]. Thus, it is possible that echinoderm larvae could share patterning mechanisms with other animals as well. Larval bisection along the AP axis produces two halves that each regenerate their missing structures. Interestingly, echinoderm larvae have been observed to undergo spontaneous cloning, highlighting a potential link to their regenerative ability and natural asexual reproduction [46].

Studying regeneration in the larvae presents numerous technical advantages over adults. Regeneration happens much more quickly in the larvae, making experimental turnover faster. The larvae are experimentally tractable, with hundreds of thousands of larvae being obtained from one fertilization. This allows for experiments with a large sample size and the collection of large amounts of material for high-throughput experiments, such as RNAseq. Larval care is simple, and large cultures can be maintained relatively easily [47].

Compared to their adult counterparts, visualization of regeneration and developmental processes and the ease with which the process can be studied molecularly is greater in the larvae [18,38,39]. Regeneration is a dynamic process and being able to observe it through, e.g., tissue structure changes and cell migration, is incredibly powerful. This information can inform how cells respond to major tissue damage and provide material to reform lost structures. Additionally, knowing precisely the source and destination of new cells for regeneration is key to understanding how this process is achieved. As the larvae are optically transparent, echinoderms are amenable to a wide variety of fluorescent microscopy studies, including lineage tracing, fluorescent in situ hybridization, and immunofluorescence. Observation of active wound healing and cell migration through time lapse imaging can also be achieved.

## 4. History of Using Echinoderm Larvae in Regeneration Studies

For over 100 years, researchers have used echinoderms to determine principles of embryonic development and regeneration. Thomas Hunt Morgan, a pioneer in genetics and embryology, took an interest in several animal models of regeneration, including sea urchins, sea stars, and brittle stars [48]. Morgan noted the regenerative capacity of the members of the echinoderm phyla, making diagrams of adult sea star arm regeneration as well as discussing the importance of where the animals were amputated and their regenerative success [49]. Beyond adult regeneration, Morgan understood the power of using sea urchin and sea star embryos to probe regeneration. Echoing earlier work, he found that cutting a sea urchin embryo was able to produce two fully developed larvae. From examinations of embryonic and adult echinoderm plasticity and regeneration, Morgan concluded that there is merit in understanding both, saying “*as there are many similarities in the two cases, and as the same factors appear in both, we cannot refuse, I think, to consider all the results from a common point of view*”. [49].

Following Morgan, countless researchers throughout the 20th century have studied regeneration utilizing echinoderm larvae (reviewed in [34]). These explorations saw the removal of particular larval tissues as well as examinations of whole-body regeneration in several echinoderm species, noting regenerative capabilities across the echinoderm tree. Of note was the work from James McClintock’s group at the University of Alabama. Using larvae from a variety of different echinoderm species, this group made important contributions to the understanding of the morphology of larval regeneration [50,51]. Following bisection, they documented the regeneration of major larval structures, including the gut, mouth, and coelomic pouches. Additionally, they noted the presence of mesenchymal cells at the wound site, hinting at the presence of a blastema-like structure [51]. They were also among the first to identify differentially expressed genes during larval echinoderm regeneration [50]. Their findings led the way for further molecular studies on the regeneration of larval echinoderms.

## 5. Genomics and Imaging: Echinoderm Tools for Understanding Regeneration

Since the sequencing of echinoderm genomes including the purple sea urchin (*Strongylocentrotus purpuratus*), the bat star (*Patiria miniata*), and other echinoderm species [52,53], there has been a wealth of genomic information and tools made available to study the molecular workings of echinoderm larval development. Thanks to techniques such as next generation sequencing and gene expression perturbation, we now have a better understanding of the gene regulatory networks that drive the formation of larval tissues and structures [54,55,56]. Having these tools available, researchers have begun to probe how these structures are reformed during regeneration [41,57]. Additionally, we can use the information gleaned from echinoderm regeneration to make comparisons to other regenerative model species.

In the last decade, larval echinoderm regeneration research has remained primarily in the sea star. While the work of McClintock’s group was extremely valuable in understanding the morphological features of larval regeneration, further work was needed to understand how this process occurs on a genetic and molecular level. One of the first dives into molecular features of larval sea star regeneration used the bat star *Patiria miniata* [57]. Previous work had identified genes as being differentially expressed during larval sea star regeneration [50], but their spatial expression had not been explored. The Wessel lab’s work examined the expression of several genes, including Vasa, a gene known to be expressed in proliferative cells in other regenerating animals [58] as well as damaged larval sea urchin arms [59]. Furthermore, utilizing the transparent feature of the larvae, they also examined cell proliferation by staining S-phase cells using the thymidine analog EdU in order to identify potential changes in proliferation levels during regeneration. This work began our molecular understanding of echinoderm larval regeneration and utilized the various tools obtained from decades of work using these animals in developmental studies.

The work of Cary et al., 2019 [41] sought to identify commonalities in regeneration among different invertebrate species (Figure 2). They generated a de novo regeneration transcriptome across three time points in both regenerating larval halves to uncover differentially expressed genes during regeneration [41]. Utilizing this dataset, they clustered the genes based on their expression profile and compared these data with regenerating transcriptomes from planaria and Hydra. From their analysis, the group uncovered key processes that underlie regeneration in sea star larvae that have been described previously in these other species. Several MAPK signaling pathway-associated genes were found to be upregulated at their earliest regeneration time point, a common pathway shown to be activated during the wound response. Having identified the change in expression of genes expressed along the AP axis following bisection, Cary et al., 2019 showed the recovery in expression of anteriorly expressed genes in posterior halves and of posteriorly expressed genes in anterior halves. This suggested that sea star larvae reestablish their AP axis during regeneration. Additionally, they observed the emergence of a proliferative blastema coinciding with an upregulation of cell cycle-associated genes later in regeneration. However, the identity and gene expression of these proliferating cells was not determined. Lastly, similarly to planarians, the sea star larvae re-proportioned their body size as regeneration proceeded. Together, this work helped pinpoint similarities among different regenerating invertebrate species and describe regenerative events in an echinoderm larvae to a depth not seen previously.

From this work on larval sea star regeneration emerges numerous biological questions that this system is well suited towards answering. Echinoderm embryos are amenable for perturbation studies using small molecule inhibitors, with some available against Wnt [60], MAPK [61], and Delta-Notch signaling [62,63]. In terms of genetic perturbation, work has begun using vivo-morpholinos in echinoderm embryos to allow for specific inhibition of gene expression through suppression of gene splicing or translation [64]. Tracing the lineage of cells during regeneration is a powerful tool for understanding the source of cells towards the reformation of new tissues. Transient transgenic echinoderm larvae have been used to mark particular lineages during development using bacterial artificial chromosome (BAC) knock-in technology [55,65]. Using these BACs during regeneration will aid in determining which populations of cells contribute to regeneration in this system. Ultimately, this information will allow for the deduction of the molecular mechanisms that drive tissue regeneration.

## 6. Conclusions

Despite being famous for their regenerative prowess, echinoderms remain understudied in the field of regenerative biology. These animals have conventionally been used in the study of embryonic development and as such, valuable genomic information and tools are available for use in their larval stage. While adult echinoderms represent an excellent model to query regenerative mechanisms in this phylum, echinoderm larvae offer a highly tractable and advantageous alternative. Beginning with morphological observations, work utilizing larval echinoderms in regeneration has evolved with the availability of new resources and tools. More insights into metazoan regeneration wait to be discovered in this new system.

## Figures and Tables

**Figure 1 genes-12-01063-f001:**
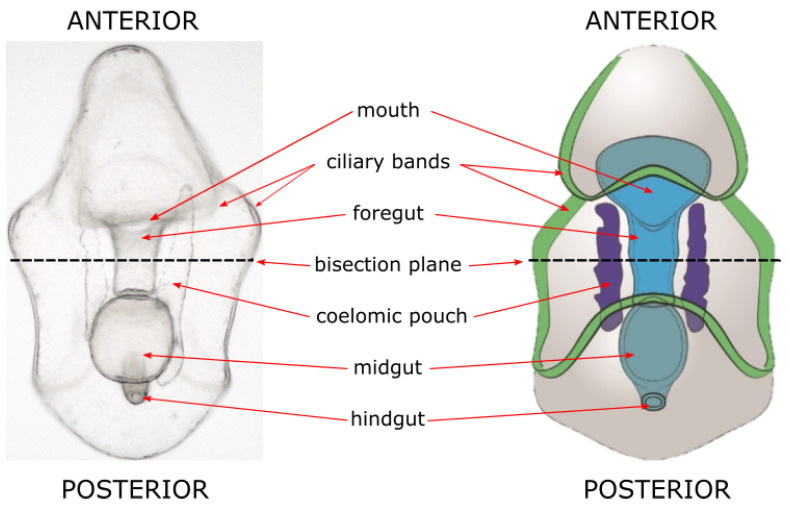
Sea star bipinnaria larval morphology. **Left**, a light microscopy image of a larva of a sea star (*Patiria miniata*). Large structures are easy to identify and tissues are transparent, making both light and fluorescent microscopy advantageous. **Right**, a schematic of major sea star larval structures.

**Figure 2 genes-12-01063-f002:**
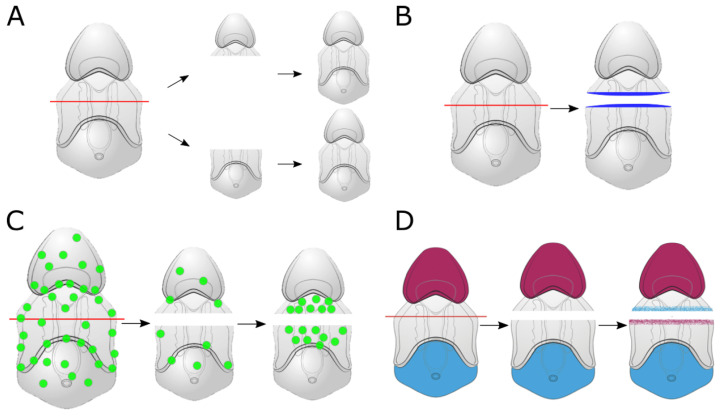
(**A**) graphical summary of the results from Cary et al. (2019) [41] examining mechanisms of larval sea star regeneration. Sea star larvae proportion their body size following bisection. This results in regenerated larvae that have similar proportions to that of an uncut larva but a smaller overall size. (**B**) Soon after bisection, the larval epithelium wound closes and expression of wound-induced genes is seen at the site of damage (blue area). This includes the expression of genes such as Elk and Egr and the localization of phosphorylated-ERK. (**C**) Subsequent to a reduction in overall cell proliferation early in regeneration (green circles), a cluster of proliferative cells emerges specifically at the injury site, akin to a regeneration blastema. (**D**) Loss of expression of genes along the anterior-posterior (AP) axis is recovered as regeneration proceeds. Many of these genes are components of the canonical Wnt-signaling pathway.

## Data Availability

Not applicable.

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
