# Peer review of "The Use of Larval Sea Stars and Sea Urchins in the Discovery of Shared Mechanisms of Metazoan Whole-Body Regeneration"

_genes, 2021, doi:10.3390/genes12071063_

Round 1

Reviewer 1 Report

The authors present a timely short review on the use of echinoderm larvae to study regeneration mechanisms.  For this, they present the work that has already been done as well as the strengths of the system.  The review is well-written and I only found minor issues that can be easily fixed.

  1. The paragraph on page 4 (lines 135-144 where the authors underscore the ease of visualization in echinoderm larvae could be strengthen by referencing some of the works where such techniques have been used, even when their use has not been focused on regenerative studies.

  1. Line 146- The term “echinoderm animals” sounds strange.  Maybe just use “echinoderms”.

  1. Line 198- When authors use the name of the authors in the text (such as Cary et al. 2019) they should still put the reference number since the references are not presented in alphabetical order.

  1. Figure 1 legend should be expanded. At least explaining that figure represents a light microscope photo, and include the species.  It could also be used to highlight the visualization issues presented by the authors.

Author Response

Point 1: The paragraph on page 4 (lines 135-144 where the authors underscore the ease of visualization in echinoderm larvae could be strengthen by referencing some of the works where such techniques have been used, even when their use has not been focused on regenerative studies.

Response 1: A few references were added to this paragraph. The work referenced uses echinoderm larvae for visualization of gene expression and cell morphology, as discussed in the text of the paper.

Point 2: Line 146- The term “echinoderm animals” sounds strange.  Maybe just use “echinoderms”.

Response 2: This sentence was corrected.

Point 3: Line 198- When authors use the name of the authors in the text (such as Cary et al. 2019) they should still put the reference number since the references are not presented in alphabetical order.

Response 3: The reference number was included in this sentence.

Point 4: Figure 1 legend should be expanded. At least explaining that figure represents a light microscope photo, and include the species.  It could also be used to highlight the visualization issues presented by the authors.

Response 4: The legend for Figure 1 has been expanded to include the points noted by the reviewer.

Reviewer 2 Report

This is a concise review of our current understanding of larval echinoderm regeneration. Given the fact that very little is known, the review covers a limited ground. However, the manuscript is well written and clear.

I have just a few (minor) comments that the authors should address:

  1. In line 68, the authors mention “sea urchin genome structure has highlighted…”. I guess that it is more appropriate to use “sea urchin genome sequence has….”. Later on, in the same line, the authors use, more properly, the expression “genomic structure”.
  2. In line 82 the authors mention “larvae are developmentally proximal”. What does this mean?
  3. If possible, rephrase the last words of the sentence in lines 86-87, “still poorly known” to avoid some word repetitions. Suggestion: “still poorly understood”.
  4. In line 122d, the authors mention that some developmental programs in echinoderm larvae are not known in adult animals. This is not the case of ophiuroid skeletogenesis where the group of Paola Oliveri have shown recapitulation of embryonic processes in the adult body (see references at the end)
  5. In figure 2 and in the text, the authors mention the presence of a blastema-like region in regenerating starfish larvae. Can you explain a bit more what are those proliferating cells? Are these stem-like cells? Do they express any stem cell-like markers?

Czarkwiani A, Ferrario C, Dylus DV, Sugni M, Oliveri P. Skeletal regeneration in the brittle star Amphiura filiformis. Front Zool. 2016 :13:18. doi: 10.1186/s12983-016-0149-x.

Czarkwiani A, Dylus DV, Carballo L, Oliveri P. FGF signalling plays similar roles in development and regeneration of the skeleton in the brittle star Amphiura filiformis. Development. 2021: 148(10):dev180760. doi: 10.1242/dev.180760.

Author Response

Point 1: In line 68, the authors mention “sea urchin genome structure has highlighted…”. I guess that it is more appropriate to use “sea urchin genome sequence has….”. Later on, in the same line, the authors use, more properly, the expression “genomic structure”.

Response 1: The confusion in this sentence has been corrected, per the reviewer’s suggestion.

Point 2: In line 82 the authors mention “larvae are developmentally proximal”. What does this mean?

Response 2: An explanation of this term has been included in the text, in the sentences where it is mentioned.

Point 3: If possible, rephrase the last words of the sentence in lines 86-87, “still poorly known” to avoid some word repetitions. Suggestion: “still poorly understood”.

Response 3: The suggested revision to this sentence was implemented.

Point 4: In line 122d, the authors mention that some developmental programs in echinoderm larvae are not known in adult animals. This is not the case of ophiuroid skeletogenesis where the group of Paola Oliveri have shown recapitulation of embryonic processes in the adult body (see references at the end)

Response 4: The suggested two references given by the reviewer were included to highlight that there is some work that has determined the relationship between developmental and adult processes in echinoderms.

Point 5: In figure 2 and in the text, the authors mention the presence of a blastema-like region in regenerating starfish larvae. Can you explain a bit more what are those proliferating cells? Are these stem-like cells? Do they express any stem cell-like markers?

Response 5: The text has been updated to address these cells. The paper this data comes from does not identify if these proliferating cells are indeed stem cells, and gene expression in these specific cells was not explored.